



# Quantifying Albedo Susceptibility Biases in Shallow Clouds

Graham Feingold[1], Tom Goren[2], and Takanobu Yamaguchi[1,3]

[1]National Oceanic and Atmospheric Administration (NOAA), Chemical Sciences Laboratory, Boulder, Colorado, USA
[2]University of Leipzig, Leipzig, Germany
[3]Cooperative Institute for Research in Environmental Sciences (CIRES), University of Colorado, Boulder, Colorado, USA

**Correspondence:** Graham Feingold (graham.feingold@noaa.gov)

**Abstract.** The evaluation of radiative forcing associated with aerosol-cloud interactions remains a significant source of uncertainty in future climate projections. The problem is confounded by the fact that aerosol particles influence clouds locally, and that averaging to larger spatial and/or temporal scales carries biases that depend on the heterogeneity and spatial correlation of the interacting fields and the non-linearity of the responses. Mimicking commonly applied satellite data analyses for calcula-

tion of albedo susceptibility $S_o$, we quantify $S_o$ aggregation biases using an ensemble of 127 large eddy simulations of marine stratocumulus. We explore the cloud field properties that control this spatial aggregation bias, and quantify the bias for a large range of shallow stratocumulus cloud conditions manifesting a variety of morphologies and range of cloud fractions. We show that $S_o$ spatial aggregation biases can be on the order of 100s of percent, depending on methodology. Key uncertainties emanate from the typically applied adiabatic drop concentration $N_d$ retrieval, the correlation between aerosol and cloud fields, and the

extent to which averaging reduces the variance in cloud albedo $A_c$ and $N_d$. Biases are more often positive than negative. $S_o$ biases are highly correlated to biases in the $\mathcal{L}$ adjustment. Temporal aggregation biases are shown to offset spatial averaging biases. Both spatial and temporal biases have significant implications for observationally based assessments of aerosol indirect effects and our inferences of underlying aerosol-cloud-radiation effects.

## 1 Introduction

Shallow liquid clouds are a poorly quantified component of the climate system and one of the greatest sources of uncertainty for climate projections (e.g. Bony and Dufresne, 2005; Bony et al., 2017). The problem is multifaceted and encompasses fundamental understanding of how these clouds are affected by the thermodynamic structure of the atmosphere, how they might change in a warmer world, how they are influenced by the atmospheric aerosol, and how all of these components are

represented in climate models. The difficulty in quantifying the radiative effects of shallow clouds emanates, to a large extent, from the large range of spatiotemporal scales involved: aerosol-cloud interaction processes need to be understood and resolved at the scale of centimeters (e.g. Hoffmann et al., 2019) while cloud fields and their organization are driven by larger scale circulations at scales of 100s to 1000s of kms (e.g. Norris and Klein, 2000). Importantly, aerosol-cloud interactions acting at





scales on the order of 100 m need to be resolved; (a) because they can lead to fundamental changes in the radiative state of
a cloud system by changing the cloud albedo, cloud fraction, and spatial distribution of condensate (e.g. Sharon et al., 2006;
Stevens et al., 2005; Wang and Feingold, 2009); and (b) because non-linearities in aerosol-cloud-radiation interactions mean
that the methodology of averaging small-scale properties to larger-scales might generate biases in the radiative response.

This paper focuses on the ramifications of small-scale processes for cloud albedo susceptibility and cloud liquid water path
adjustments (changes in liquid water path in response to changes in aerosol concentration) within the context of how they are
treated in satellite-based analyses. Two key aspects are addressed: the first relates to spatial averaging from the level of the
satellite pixel (order 1 km) to commonly used aggregation scales on the order of 10s – 100s of kms; the second relates to
temporal averaging from the individual scene snapshot up to a timeframe on the order of months. Large scale analyses often
aggregate data both spatially and temporally into data sets that might cover e.g. $4° \times 4°$ and five years (Chen et al., 2014), or
a range of scales from kms - 10s kms for cloud microphysics, liquid water path, and radiation, $1° \times 1°$ for aerosol, and three
month seasonal responses (Lebsock et al., 2008). While it is impractical not to aggregate to some degree – e.g., to smooth noisy
retrievals or extract signals from a noisy background – the implications for quantifying aerosol-cloud interactions are still not
well understood.

We quantify aerosol-cloud interactions using the albedo susceptibility metric (Platnick and Twomey, 1994) defined here as

$$S_o = \frac{dlnA_c}{dlnN_d} = \frac{(1 - A_c)}{3} \left[ 1 + \frac{5}{2} \frac{dln\mathcal{L}}{dlnN_d} \right], \tag{1}$$

where $A_c$ is the cloud albedo, $N_d$ is the drop concentration, and $\mathcal{L}$ is the liquid water path. The expression comprises the cloud
brightening or 'Twomey' component $(1 - A_c)/3$ and the adjustment term $\mathcal{L}_o = dln\mathcal{L}/dlnN_d$, and assumes no changes in drop
distribution width (e.g. Feingold and Siebert, 2009). The factor of 5/2 suggests a potentially strong but uncertain contribution
from $\mathcal{L}$ adjustments; even the sign of this term varies from positive for precipitating clouds to negative for non-precipitating
clouds (Christensen and Stephens, 2011; Glassmeier et al., 2021).

Using satellite-based observation systems, e.g., the MODerate Imaging Spectroradiometer (MODIS; Salomonson et al.,
1998), one can derive a drop concentration $N_{d,a}$, based on adiabatic assumptions, from retrieved visible cloud optical depth $\tau$
and cloud-top drop effective radius $r_e$:

$$N_{d,a} = \frac{\sqrt{5}}{2\pi k} \left( \frac{f_{ad} \, c_w(T,P) \, \tau}{Q_{ext} \, \rho_w \, r_e^5} \right)^{1/2}, \tag{2}$$

where $c_w(T,P)$ is related to the condensation rate and is a know function of cloud-base temperature $T$ and pressure $P$, $f_{ad}$
is the adiabatic fraction (assumed in this paper to be 0.8), $Q_{ext}$ is the extinction coefficient ($\approx 2$ in the visible part of the
spectrum), $\rho_w$ is the density of liquid water and $k$ is a factor that is inversely proportional to the width of the drop size
distribution (assumed to be 0.8). When $f_{ad} = 1$, $N_{d,a}$ is the adiabatic drop concentration.

Liquid water path $\mathcal{L}$ is derived from MODIS data as

$$\mathcal{L} = \frac{5}{9} \, f_{ad} \, r_e \, \tau. \tag{3}$$



Further details of these derivations can be found in Brenguier et al. (2000) and Grosvenor et al. (2018). $A_c$ can be derived from $\tau$ using a simple two-stream approximation for a plane-parallel cloud (Bohren, 1987)

$$A_c = \frac{\tau}{\gamma + \tau} \tag{4}$$

where $\gamma$ depends on the degree of forward scattering. Equation(4) also assumes an overhead sun, no absorption, and a dark underlying surface. We do not consider 3-dimensional radiative transfer.

As an example of how aggregation of data can affect the quantification of derived variables, we note that Eq. (4) is a concave function, which following Jensen's inequality, means that for an inhomogeneous cloud field, $f(\bar{\tau}) > \overline{f(\tau)}$. Thus calculating $\tau$ based on large length scale-averaged cloud properties, and then calculating $A_c = f(\bar{\tau})$ using Eq.(4), will generate a high bias in $A_c$ that is inherently a function of the inhomogeneity of the cloud field. Because this well-known albedo bias (e.g. Cahalan et al., 1994; Oreopoulos and Davies, 1998) is not the topic of this paper, we will assume in all calculations that $A_c$ is

measured directly by an instrument like Clouds and the Earth's Radiant Energy System (CERES) at the desired measurement length-scale, and therefore does not suffer aggregation bias. Instead, we explore similar biases that affect quantification of $S_o$. For example, Eq. (2) is a highly non-linear function of $\tau$ and $r_e$ so that whether one elects to calculate $N_{d,a}$ before or after averaging of component variables $\tau$ and $r_e$ will potentially have a strong effect on $S_o$.

McComiskey and Feingold (2012) analyzed large eddy simulation (LES) output of three cloud fields characterized by differ-

ent degrees of inhomogeneity and showed that an aerosol-cloud interaction metric $\mathrm{dln}\tau/\mathrm{dln}N_d$ increased as a result of averaging – more so for heterogeneous fields than for homogeneous fields. To put the topic on firmer footing we first establish a theoretical framework for assessing the biases. We then use a large number of LESs (127) as proxy data, which we then use to simulate satellite retrievals. We apply typical methodologies used in satellite retrievals, as well as variants, to assess the provenance of the $S_o$ bias. Both spatial and temporal aggregation scales are considered, with a focus on the former. Because of the limited

domain size available to LES, we concern ourselves with the effects of spatial averaging from scales on the order of 1 km to 10 km. The multiple snapshots of different scenarios available from the LES output provide the basis for temporal aggregation.

## 2   Theory

### 2.1   Spatial aggregation of variables derived from non-linear functions

The two fundamental geophysical variables associated with aerosol-cloud interactions are $N$ (a generic concentration; drop

or aerosol number, which are well-correlated) and $\mathcal{L}$. In the case of homogeneous aerosol and cloud fields, aggregation of data to different scales has no effect on derived quantities such as $N_d$, $\mathcal{L}$, $A_c$, and $S_o$, and the order of calculation of these fields is of no consequence. In reality, however, cloud fields exhibit different degrees of inhomogeneity: Condensation of cloud water responds to local updrafts, and to some extent, availability of cloud condensation nuclei. Drop concentration depends on aerosol concentration – typically a less variable field than cloud water – as well as local supersaturation driven by updrafts.

Under these conditions, the quantification of aerosol-cloud interaction metrics like $S_o$ and the influence of averaging could be far more important.





The theoretical framework for addressing this question is well-known from similar examples in the atmospheric sciences, notably biases in rain formation processes that result from large-scale averaging of the cloud water and drop concentration terms in expressions for autoconversion and accretion (e.g. Lebsock et al., 2013; Zhang et al., 2019). In the interests of completeness,

we repeat the key equations here. Assuming a lognormal PDF of quantity $x$:

$$P(x) = \frac{1}{\sqrt{2\pi}\, x\, ln\sigma_{g,x}} exp[-(x - x_g)/(2ln^2\sigma_{g,x})] \tag{5}$$

where the lognormal parameters are geometric mean (or median) $x_g$ and geometric standard deviation $\sigma_{g,x}$. Quantity $x$ represents $A_c$ or $\mathcal{L}$, and $N$. Using well known integral properties of the lognormal function (e.g. Feingold and Levin, 1986) it is easy to show that the bias in a moment $x^{\beta_x}$ associated with averaging can be written as:

$$B_x = \frac{\overline{x^{\beta_x}}}{\overline{x}^{\beta_x}} = (D_x^2 + 1)^{\frac{\beta_x^2 - \beta_x}{2}} \tag{6}$$

with the relative dispersion $D_x$, the ratio of standard deviation to the mean, defined as

$$D_x = (\overline{(x - \overline{x})^2})^{1/2}/\overline{x}. \tag{7}$$

If the two interacting fields, e.g. $\mathcal{L}$ and $N$, are assumed to follow a bivariate lognormal distribution, the bias associated with the covariance between $\mathcal{L}$ and $N$ is

$$B_{cov} = exp(r(\mathcal{L}, N) \cdot \beta_\mathcal{L} \cdot \beta_N \cdot \sigma_{g,\mathcal{L}} \cdot \sigma_{g,N}). \tag{8}$$

where $r(\mathcal{L}, N)$ is the spatial correlation between $\mathcal{L}$ and $N$. (Note that we elect to present the theory in terms of $\mathcal{L}$ and $N$ rather than $A_c$ and $N$ because $A_c$ includes compounded dependence on $N$ via $\tau$. Since $\mathcal{L}$ and $A_c$ are highly correlated, this choice does not affect the general framework for discussion.)

The overall bias associated with averaging for two covarying fields is then given by

$$B = B_\mathcal{L} \cdot B_N \cdot B_{cov}. \tag{9}$$

The equations allow theoretical calculation of the $S_o$ biases associated with two covarying fields $\mathcal{L}$ and $N$, each characterized by its own heterogeneity $D_\mathcal{L}$ and $D_N$, respectively. These are shown in Fig. 1 in $D_\mathcal{L}$; $D_N$ space, for three values of $r(\mathcal{L}, N)$. We note that large positive and negative biases can result from averaging; for negative $r(\mathcal{L}, N)$, biases are positive, and on the order of $20 - 60$ % whereas for positive values of $r(\mathcal{L}, N)$, biases are negative, and on the order of -20 – -70%. When

correlation between the fields is zero, biases are $10 - 20$ %. The central role of $r(\mathcal{L}, N)$ is further illuminated by plotting the bias in $S_o$ as a function of $r(\mathcal{L}, N)$ for specified $D_\mathcal{L}$ and $D_N$ combinations (Fig. 2).

Note that the assumption of a bivariate lognormal distribution is common when dealing with geophysical fields. Regardless of how well our fields actually adhere to this assumption, the theoretical analysis does provide a framework within which to identify possible key controls over biases.




## 2.2 Effects of spatial and temporal aggregation on variance, and correlation between fields

The second framework for assessment of biases derives from the basic definition of the linear regression fit:

$$\hat{b} = r_{x,y}\frac{\sigma_y}{\sigma_x} \tag{10}$$

where $\hat{b}$ is the regression slope and $\sigma_x$ is the standard deviation of field $x$. In our case, $x$= aerosol or drop concentration (we will use $N_d$) and $y$ = cloud variable ($A_c$ or $\mathcal{L}$). As shown by McComiskey and Feingold (2012), aggregation increases $r(x,y)$ but decreases $\sigma_x$ and $\sigma_y$ to varying degrees. Of interest is therefore, the extent to which aggregation changes $r(x,y)$ and the ratio $\sigma_y/\sigma_x$.

## 3 Large Eddy Simulation as a Data Source

We calculate $S_o$ biases using output from 127 large eddy simulations of marine stratocumulus under a range of conditions from fairly homogeneous overcast to broken open cellular structures.

Simulations are generated by the System for Atmospheric Modeling (SAM; Khairoutdinov and Randall, 2003). Input conditions are derived from ERA-5 reanalysis in the stratocumulus regime off the coast of California. The model set-up is similar to Feingold et al. (2016) and Glassmeier et al. (2021), with initial meteorological and aerosol conditions sampled using the Latin hypercube method. Simulations are nocturnal and of 12 h duration. The first 5 h of output from each simulation is discarded to avoid cloud scenes that are not fully developed. The domain size is 48 km × 48 km × 2.5 km with a 500 m damping layer below the model top at 2.5 km. The profile is extended up to 35 km (5 hPa) for radiation calculations. Grid spacings are dx = dy = 200 m, dz = 10 m. A two-moment bulk microphysical method that calculates all warm cloud processes (including supersaturation and activation) is applied (Feingold et al., 1998). The simulations differ from Glassmeier et al. (2021) in one key respect, namely surface fluxes are calculated interactively. Resultant cloud fields exhibit varying degrees of heterogeneity, including closed-, open-, and transitions from closed-to-open cellular convection.

The LES output provides microphysical fields drop number concentration and liquid water content as three-dimensional prognostic variables, under the assumption of a bimodal lognormal distribution of cloud drops and rain drops with fixed distribution width ($\sigma_g$ = 1.2 for both). However to mimic satellite retrievals, we work with the derived model variables $\tau$ and cloud-top $r_e$ to calculate $N_d$, and $\mathcal{L}$ based on Equations (2) and (3). Both cloud water and rain water contribute to $\tau$ and $r_e$. $A_c$ is calculated based on the modeled value of $\tau$ (Eq. 4) and then averaged, and therefore does not suffer from aggregation bias. $S_o$ is calculated directly for each scene based on the definition ($S_o = dlnA_c/dlnN_d$) using least squares regression to the natural logarithms of $A_c$ and $N_d$.





### 3.1 Spatial Aggregation

#### 3.1.1 Level 2 Analysis

The standard averaging method follows the MODIS level 2 (L2) methodology in the sense that calculations are performed at
high resolution prior to averaging. In the current work, Eqns. (2) and( 3) are calculated based on the native 200 m model mesh
($n =1$) and then aggregated to $n \times n$ tiles. Results will be shown for $n = 30$ vs. $n = 4$, i.e., 6 km $\times$ 6 km boxes vs. 800 m $\times$ 800
m boxes. These choices result from a desire for a reasonable number of regression points ($n = 30$) and for some small amount
of smoothing ($n = 4$) to reduce the noise in $N_d$ retrievals (Eq. 2). The choice of $n = 4$ also brings us close to the typical 1 km
length-scale used for analysis of L2 MODIS data. Note that L2 methodology removes the biases associated with aggregation
of non-linear functions discussed in section 2.1 (Jensen's inequality) since $N_d$ is calculated using Eq. (2) at the pixel level and
then averaged up. The same is true for $A_c$, which as previously noted is calculated based on Eq. (4).

Biases are defined as

$$(\bar{X} - X)/X, \tag{11}$$

where X represents $S_o$ calculated at $n = 4$ and $\bar{X}$ represents $S_o$ at $n = 30$ . Correlations $r(\mathcal{L}, N_d)$ refer to calculations for $n = 4$
unless otherwise stated. $N_d$ in the L2 analysis represents cloudy column averaged drop concentration, i.e., $N_d = N_{d,a}$ based
on Eq.(2).

The $N_d$ retrieval (Eq. 2) is very sensitive to thin clouds particularly when $r_e$, but also $\tau$ are small. As is common in MODIS
data analyses, calculations are only applied to thicker clouds, in our case, to $r_e > 3$ $\mu$m and $\tau > 3$.

#### 3.1.2 Variants

Two variants of the calculations will be presented.

1. Mimic MODIS level 3 (L3) analysis. Here satellite-based retrievals are based on aggregated data. In other words Eqns.
   (2) and (3) for $N_d$ and $\mathcal{L}$, respectively, are applied to data aggregated to $n = 4$ and $n = 30$. By doing so the aggregation
   biases associated with Jensen's inequality are introduced. The aggregation biases are expected to derive from a mix of
   influences: low for $N_d$ (a convex function in $r_e$ dominates a concave function in $\tau$; Eq. 2), and negligible for $\mathcal{L}$ (Eq.
3). The effect of smoothing associated with level 3 aggregation is thus likely to be highly dependent on cloud field
   heterogeneity.

2. Mimic MODIS L2 analysis but eliminate uncertainties in the (sub)adiabatic retrieval of $N_d$ by assuming a 'perfect' $N_d$,
   which is taken directly from the LES. Because the LES $N_d$ is a 3-dimensional variable, we use in-cloud column average
   values. This methodology will be referred to as L2$_{\mathrm{N}}$. Note that $\mathcal{L}$ is still calculated as per Eq. (3). The goal here is to
understand the extent to which the retrieval of $N_d$ drives the regression biases.





### 3.2 Temporal aggregation

To address temporal aggregation we consider the same individual LES snapshots described above but calculate $S_o$ in two ways: (i) $S_o$ regression fits to individual cloud scenes are simply averaged up over all cloud scenes; (ii) LES output is temporally aggregated to a large data set, from which $S_o$ is calculated via regression. The first approach preserves the individual cloud

scene susceptibilities while the second aggregates many different cloud fields before performing the regression. The biases are calculated based on Eq. (11), with the overbar indicating the second approach. The methodology is followed (separately) for both $n = 4$ and $n = 30$ spatial aggregation, and for L2, L3, and L2$_\mathrm{N}$.

## 4   Results

### 4.1   Spatial Aggregation

#### 4.1.1   Effect of Spatial Aggregation on $S_o$ and correlation

Figure 3 presents results for the three approaches (L2, L3, and L2$_\mathrm{N}$). We start with the figures showing $\bar{S}_o$ vs. $S_o$ to avoid ambiguity in the sign of the bias associated with Eq. 11. (According to Eq. (11), conditions under which $\bar{S}_o$ is more negative than $S_o$ also manifest as positive biases.) The solid line is the 1:1 line.

Points are colored by cloud fraction $f_c$ since it also serves as a good proxy for cloud field heterogeneity. (The correlation

between $f_c$ and $D(\mathcal{L})$ is 0.86.) For L2, we note that both $\bar{S}_o$ and $S_o$ are almost always positive and that low $f_c$ states do not suffer from worse biases than high $f_c$ states. On the contrary, high $f_c$ is often associated with the largest biases.

Responses for the L3 analysis are distinctly different in a number of ways: first, both $\bar{S}_o$ and $S_o$ are almost always negative, and low $f_c$ cloud scenes often have lower biases than high $f_c$ scenes. This is because L3 averaging has a stronger smoothing effect on broken cloud fields, and therefore somewhat unexpectedly *reduces* the averaging bias for broken cloud fields com-

pared to solid cloud fields. Nevertheless, the non-physical shift in the sign of $S_o$ associated with L3 methodology should act as a cautionary note.

L2$_\mathrm{N}$ analysis yields strongly positive $S_o$ and a clearer dependence of the bias on $f_c$. For broken cloud scenes $S_o$ is sometimes negative but biases tend to be scattered and relatively small. These low $f_c \approx 0.3$ states are dominated by cumulus cells with stronger updrafts that result in coherent $N_d$. Since $S_o$ and $\bar{S}_o$ are only calculated in cloudy regions (above the $r_e$ and $\tau$

thresholds), this coherence in $N_d$ results in a small bias. Thus the reasons for small susceptibility bias at low $f_c$ differ for L3 and L2$_\mathrm{N}$. With increasing $f_c$, biases tighten around the 1:1 line but start to deviate for $f_c > 0.85$, and exhibit increasingly large values.

To quantify the biases, these same analyses are shown as % biases in $S_o$ (Eq. 11) as a function of the correlation between $\mathcal{L}$ and $N_d$ ($r(\mathcal{L}, N_d)$; Fig. 4), the calculations of which are consistent with the derivation of variables averaged to $n = 4$; e.g., L2

and L3 calculations apply $r(\mathcal{L}, N_d)$ based on Eqns. 2 and 3, and L2$_\mathrm{N}$ calculates $r(\mathcal{L}, N_d)$ using the true $N_d$ and Eq. 3.



L2 biases are almost always positive and can reach values of many 100s of % (Fig. 3). As expected from Section 2.1, $r(\mathcal{L}, N_d)$ has a strong influence over the $S_o$ bias, particularly for L2, with the bias increasing noticeably with decreasing $r(\mathcal{L}, N_d)$, in a manner qualitatively similar to Fig. 2. The high values and high variability in the bias as one approaches $r(\mathcal{L}, N_d) \approx 0$ are to some extent a consequence of an uncertain regression fit when the correlation between the $\mathcal{L}$ (or the closely related $A_c$) and $N_d$ fields is poor.

Of note is that L3 analysis methodology (aggregate first, then derive) changes the sign of $r(\mathcal{L}, N_d)$ to negative values, as it did the sign of $S_o$ (Fig. 3). L3 biases are more widely dispersed and show no clear trend with $r(\mathcal{L}, N_d)$ or $f_c$. L2$_N$ biases tend to be capped at about 100% and values of $r(\mathcal{L}, N_d)$ are noticeably more positive than those in L2. These results reinforce two points: (i) that L3 analysis generates non-physical results (negative $S_o$ and a change in the sign of $r(\mathcal{L}, N_d)$); and (ii) that the use of the (sub)adiabatic assumption (Eq. 2) as a proxy for $N_d$ incurs a significant increase in the $S_o$ bias relative to L2$_N$, most noticeably at low $r(\mathcal{L}, N_d)$ where Eq. 2 results in a significantly reduced (but for the most part, positive) correlation.

### 4.1.2 Effect of Spatial Aggregation on Regression

It is useful to turn to the underlying definitions of regression analysis (section 2.2) to explore more deeply the influence of averaging on the $S_o$ bias. According to Eq. (10), $S_o$ biases are related to the ratio of $\hat{b}$ (6 km) to $\hat{b}$ (800 m):

$$\bar{\hat{b}}/\hat{b} = \frac{\bar{r}_{x,y}}{r_{x,y}} \left(\frac{\bar{\sigma}_y}{\bar{\sigma}_x}\right) / \left(\frac{\sigma_y}{\sigma_x}\right). \tag{12}$$

We therefore consider (i) the ratio $\bar{\sigma}_{A_c}/\bar{\sigma}_{N_d}$ to $\sigma_{A_c}/\sigma_{N_d}$ (henceforth the '$\sigma$-ratio'; Fig. 5), and (ii) the ratio $\bar{r}(A_c, N_d)/r(A_c, N_d)$ (henceforth the '$r$-ratio'; Fig. 6), which by Eq. (10) are both determinants of the bias. Note that for the $r$-ratio we use $r(A_c, N_d)$ to adhere more rigorously to the regression analysis definitions. The interpretation of these ratios is non-trivial. They express the extent to which $\sigma_{A_c}/\sigma_{N_d}$ and $r(A_c, N_d)$ are modified by aggregation. In the case of $\sigma_{A_c}/\sigma_{N_d}$ this amounts to interpreting a 'ratio of ratios'. Later we will delve into this in more detail.

L2 results show a clear dependence of the $S_o$ bias on the $\sigma$-ratio, and especially large biases when the $\sigma$-ratio is high (Fig. 5a). These also happen to be points exhibiting high $f_c$ (cf. Fig 4a). Also apparent is that the separation of positive and negative biases is demarcated at a $\sigma$-ratio of 1. The results suggest a strong correlation between the $\sigma$-ratio and $f_c$. At high $r(\mathcal{L}, N_d)$, the $S_o$ bias increases systematically with increasing $\sigma$-ratio (Fig. 5a) but with decreasing $r(\mathcal{L}, N_d)$ the strong, and orthogonal influence of the $r$-ratio becomes more important (Fig. 6a). Clearly evident in Fig. 6a is the anticipated unstable calculation of the $r$-ratio in the vicinity of $r(\mathcal{L}, N_d) = 0$.

The L3 methodology exhibits an even clearer dependence of the $S_o$ bias on the $\sigma$-ratio, except for cloud fields with $r(\mathcal{L}, N_d) \approx -0.2$ (Fig. 5b) where the high bias is clearly related to both the $\sigma$- and $r$-ratios (Fig. 6b). The aforementioned vertically oriented green colored points have a $\sigma$-ratio of about 1 and $r$-ratio of 2.5, i.e., the bias is driven by the $r$-ratio.

For the L2$_N$ methodology, here too the $\sigma$-ratio (Fig. 5c) provides a clearer indication of the magnitude of the bias compared to either the $r$-ratio (Fig. 6c) or $f_c$ (Fig. 4).

Based on Eq. (12), the influence of the $\sigma$- and $r$-ratios on $S_o$ bias is expected. What is more revealing is the influence of averaging on the *components* of these ratios. To this end, Fig. 7 examines the effect of averaging on the reduction in $\sigma_{A_c}$ and





$\sigma_{N_d}$. In other words, we ask: to what extent does averaging smooth the $A_c$ field relative to the smoothing in the $N_d$ field?

Accordingly, axes in Fig. 7 are calculated as normalized reductions in the variables.

L2 analysis shows that at low $f_c$ the normalized reductions in $\sigma(N_d)$ tend to be smaller than those in $\sigma(A_c)$ but that the reverse tends to be true for $f_c > 0.75$ (Fig. 7a), i.e. at high $f_c$, averaging smooths the $N_d$ field more than it smooths the $A_c$ field. We will show below that these high $f_c$ scenes, although inherently more homogeneous, often manifest significant inhomogeneity in $N_d$ as a result of the $N_d$ retrieval. (See further discussion in section 4.1.4; Examples.)

The clear exception to the trend of aggregation smoothing $N_d$ more than $A_c$ with increasing $f_c$ is the group of low $f_c$ points that exists in Fig. 4a at $r(\mathcal{L}, N_d) < 0.2$. These anomalous points appear below the 1:1 line in Fig. 7a and show up as lower $r$-ratio points (reddish points) in a sea of higher $r$-ratio points (brown colors) in Fig. 8a. Another distinct feature is the group of vertically oriented high $f_c$ (Fig. 7a) and low $r$-ratio (Fig. 8a) points for which smoothing of $N_d$ increasingly exceeds smoothing in $A_c$ as one moves below the 1:1 line. Because these points are characterized by small values of the $r$-ratio, there appears to

be an offsetting of $\sigma$-ratio and $r$-ratio effects.

A closer look shows that these points manifest as a negative $S_o$ bias (Fig. 9a); in other words, the reduction in the $r$-ratio dominates the increase in the $\sigma$-ratio. Although these negative bias points tend to be more rare, they can be identified in Fig. 4a at high $f_c$ and low $r(\mathcal{L}, N_d)$ ($< 0.3$).

Analysis of L3 shows some similarities and some differences from L2. First, there is a much more significant scatter in

points, particularly at lower $f_c$ (Fig. 7b); second, as in L2, averaging-related smoothing of $N_d$ tends to exceed smoothing in $A_c$ at higher $f_c$ (Fig. 7b); third, and different from L2, values of the $r$-ratio of $\approx 1$ (Fig. 8b, green colors) are associated with higher $S_o$ biases (Fig. 9b), which must derive from the $\sigma$-ratio.

Finally, L2$_N$ reveals a somewhat richer palette of responses. First the commonalities: (i) the general trend for smoothing of $N_d$ to exceed smoothing of $A_c$ with increasing $f_c$ and increasing $r$-ratio is relatively robust (cf. Fig 7c and Figs 7ab); (ii)

when the $r$-ratio $\approx 1$, and smoothing of the fields is similar (Figs. 8a and 8c), the $S_o$ bias is capped at about 100% (Figs. 9a and 9c). *In fact it is clear from Fig. 9 that $S_o$ biases exceeding 100% always occur when averaging-related smoothing has a stronger effect on $N_d$ than on $A_c$,* although the magnitude and even sign of these biases vary significantly depending on the methodology. The richness (lack of monotonicity) in responses under these conditions depends on the relative strength of the $r$-ratio, and the extent to which it amplifies or counteracts the $\sigma$-ratio.

We note one interesting difference between L2 and L2$_N$: for the latter, low $f_c$ points reside in conditions under which averaging-related smoothing is dominated by more *as well as less* significant smoothing of $N_d$ vs $A_c$. (Fig. 7c). In very rare cases the low $f_c$ points above, but close to the 1:1 line in Fig. 9c cause negative $S_o$ biases (cf. Fig. 6c). Finally, some very high positive $S_o$ biases (up to 500 %) do exist for L2$_N$. These can be traced to conditions when the $r$-ratio is large (Fig. 8c) and averaging smooths $N_d$ more than $A_c$. In this case the effects of averaging on the $r$-ratio and the $\sigma$-ratio work in unison to

amplify the bias.





### 4.1.3 $S_o$ bias vs. $\mathcal{L}$ adjustment bias

The topic of $\mathcal{L}$ adjustments is of great interest given that the term may both enhance or offset the overall albedo susceptibility (Eq. 1) (e.g. Glassmeier et al., 2021). For example, a value of $\mathcal{L}_o = dln\mathcal{L}/dlnN_d < -0.4$ will change the sign of $S_o$ from positive to negative. Numerous recent articles, based on models and observations, point to $\mathcal{L}_o$ being positive in precipitating

conditions, following the familiar Albrecht (1989) 'cloud lifetime' hypothesis which posits that aerosol perturbations will suppress collision-coalescence and decrease precipitation, and therefore $\mathcal{L}$ losses, while it is negative in the non-precipitating regime, as a result of enhanced evaporation-entrainment feedbacks (Wang et al., 2003; Ackerman et al., 2004; Xue et al., 2008; Christensen and Stephens, 2011; Gryspeerdt et al., 2019). Fig. 10 shows the relationship between spatial averaging-related $S_o$ and $\mathcal{L}_o$ biases, with points colored by $f_c$. For clarity we show a subsample of 58 of the total 127 simulations to avoid points

clustering and obscuring points below.

First, we note a strong positive correlation between the two biases, with $\mathcal{L}_o$ biases larger than $S_o$ biases in L2 and L2$_\text{N}$ (Fig. 10a, c). The reverse is true for L3 (Fig. 10b). For L2 and L3, two distinct branches appear; the first is a tight relationship for $f_c > 0.7$, while the second is somewhat less well defined and associated with lower $f_c$. There is a saturation in the ratio for $f_c$ on the order of 0.3 in L2 (Fig. 10a), while L3 shows a distinctly stronger $S_o$ bias at low $f_c$ compared to the approximately linear

relationship for high $f_c$ (Fig. 10b). The separation of these branches is much less distinct for L2$_\text{N}$ (Fig. 10c) but in general $\mathcal{L}_o$ biases are larger than $S_o$ biases, although to a lesser extent than in L2. While the strong positive correlation between $S_o$ and $\mathcal{L}_o$ biases is not surprising given the expected tight relationship between $S_o$ and $\mathcal{L}_o$, it is clear that using the methodologies applied here, satellite-based analyses of the $\mathcal{L}_o$ bias have the potential to be at least as severe as those associated with $S_o$ biases.

### 4.1.4 Examples

While our goal has been to provide a broad assessment of susceptibility biases in terms of cloud field properties, some examples are helpful to illustrate the issues. We focus on L2 and L2$_\text{N}$ to isolate the effect of the $N_d$ retrieval for individual cases, chosen randomly based on their visual physical characteristics, but supported by other cases. Figure 11 presents an LES-generated stratocumulus scene characterized by high $f_c$ (= 0.98) and a very realistic closed-cellular structure ( Fig. 11a,c). Drop concentration fields for the (sub)adiabaticlly-derived $N_d$ (Eq. 2) and the 'true' (LES) $N_d$ show the problem very clearly.

The retrieved $N_d$ shows a great deal more fine-scale structure than the true $N_d$, and although in the mean the retrieved $N_d$ is not highly unrealistic (retrieved $N_d = 167$ cm$^{-3}$; true $N_d = 227$ cm$^{-3}$ — an error of -26 %), the two fields are strongly negatively correlated, with a fit slope of $N_d$ (retrieved) = 378 - 0.9 × $N_d$ (true). To quantify this further, we consider the correlations between $\mathcal{L}$ and $N_d$: Applying the true $N_d$, $r(\mathcal{L}, N_d) = 0.78$ whereas using the retrieved $N_d$, $r(\mathcal{L}, N_d) = -0.24$. This shift from strong positive $r(\mathcal{L}, N_d)$ to negative $r(\mathcal{L}, N_d)$ has implications for the $S_o$ bias (E.g. Fig. 2): The L2-derived $S_o$ bias is -2017

% whereas the L2$_\text{N}$ bias is +80 %. Here the use of true $N_d$ reduces the $S_o$ error very significantly. This example serves to explain why high $S_o$ biases can exist in high $f_c$ scenes (e.g. Fig. 4). We emphasize that the significant inhomogeneity in $N_d$ is a result of the $N_d$ retrieval, rather than an inherent property of the cloud field, and that it is this increase in inhomogeneity and reduction in $r(\mathcal{L}, N_d)$ that drives up the $S_o$ bias.





The second example (Fig. 12) is a low $f_c$ case (0.23) exhibiting classic open-cellular structure. Here the mean retrieved $N_d =$

21 cm$^{-3}$ and the true $N_d = 12$ cm$^{-3}$ — an error of +75 %. The best fit linear regression between the two yields $N_d$ (retrieved) $= 20 + 0.075 \times N_d$ (true). Examining the true $N_d$, we find $r(\mathcal{L}, N_d) = 0.13$ whereas using the retrieved $N_d$, $r(\mathcal{L}, N_d) = -0.05$ The $S_o$ biases are 285 % and 803 % for L2 and L2$_N$ respectively, i.e., the true $N_d$ *degrades* the $S_o$ bias. We have identified two contributing factors to this unexpected result: (i) in the case of L2$_N$, the proximity of $r(\mathcal{L}, N_d)$ to zero generates an unstable $r$-ratio and unstable $S_o$ bias (Eq. 12); (ii) more generally, unexpected results can occur when the base $S_o$ ($n = 4$) is small,

which by Eq. (11), will generate unstable bias calculations. For the current case, $S_o$ is small because the open cell walls are already very bright, i.e., $(1 - A_c)$ in Eq. (1) is small.

While examination of case studies proves useful, we argue that the broader statistical view is essential to understanding the error landscape that might be encountered in highly variable natural cloud scenes. In this regard, Figs. 3 – 6, supported by Figs. 7 – 9, are essential.

**4.2   Temporal Aggregation**

Although the focus of the work has been on spatial aggregation, we now briefly consider the effects of temporal aggregation, which by Eq. (10) will also affect $S_o$. Large scale analyses often aggregate data over multiyear timescales (e.g. Chen et al., 2014), or three month seasonal timescales (Lebsock et al., 2008), extending the range of conditions and changing the variance and correlation between the fields. The result is that the regression fit to a longterm, temporally aggregated data set will be

different from the short timeframe fits, averaged up to include the same data.

Table 1 summarizes the results for the three averaging methodologies, and for spatial averaging of $n = 4$ and $n = 30$. Here $\Sigma S_o$ denotes a simple average of the $S_o$ fits to individual scenes while $\bar{S}_o$ aggregates all scenes before performing the regression. Because of the large number of individual $A_c; N_d$ pairs required to calculate $\bar{S}_o$, calculations are limited to 58 of the 127 simulations. Our experience has shown that these 58 are representative of the full set of 127. The biases are calculated

based on Eq. (11). Of immediate note is that the analysis shows that temporal aggregation results in a reduced $S_o$ on the order of 70 – 110 %, depending on the methodology applied. For L2 and L2$_N$ the average of local-in-time cloud albedo susceptibility is *larger* than that calculated by temporally aggregating many scenes. This bias is of opposite sign to the typical biases associated with spatial aggregation (e.g. Fig. 4) and not significantly different in magnitude. This offsetting of errors should be seen as a cautionary flag when making choices of how to aggregate data, rather than as a fortuitous occurrence, since as we have shown

above, the biases are highly dependent on cloud field properties.

In the case of L3, the temporal aggregation of many different scenes results in an increase in albedo susceptibility and a change in sign from negative to weakly positive. Here too, the spatial and temporal aggregation has opposite effects (cf. Fig. 3b). (Note that the % differences for L3 in Table 1 are somewhat misleading: because of the change in sign, increases in $S_o$ due to temporal averaging still show up as negative differences because of the normalization by the negative $\Sigma S_o$.)

Finally, given the close relationship between the $S_o$ and $\mathcal{L}_o$ spatial aggregation biases, we surmise that temporal averaging will have similar effects on $\mathcal{L}_o$ as on $S_o$.





## 5    Discussion and Summary

Satellite-based measurements are our best means of assessing aerosol-cloud-radiation interactions at the global scale and providing model constraints for one of the most uncertain forcings of the atmospheric system, namely aerosol indirect effects.

Space-based data draw heavily on polar-orbiting satellites carrying passive instruments from which we infer aerosol properties (e.g., aerosol optical depth), cloud properties (cloud optical depth, cloud-top drop effective radius, liquid water path), and radiative fluxes. Typical approaches to quantification of aerosol-cloud-radiation interactions aggregate these inferred properties spatially and temporally to suppress the 'noisy' retrievals inherent to these measurements. The goal of this paper has been to address the ramifications of spatial and temporal aggregation for standard metrics of indirect effects in the form of cloud

albedo susceptibility $S_o$, which is in turn strongly dependent on the liquid water path adjustment ($\mathcal{L}_o$) (Eq. 1). The question is central to our ability to quantify aerosol indirect effects, and raises fundamental questions of non-linearities in the aerosol-cloud interaction system, and natural co-variability of the interacting fields. Early work recognized the effects of aggregation on cloud albedo ($A_c$) – the so-called "albedo bias" (e.g. Cahalan et al., 1994). The current study has assumed no aggregation error in $A_c$ (as in the case of a direct CERES-based retrieval) but has focused instead on derivatives of the form of Eq. (1)

that include highly non-linear satellite-based retrievals of drop concentration ($N_d$). In addition, we have derived liquid water path $\mathcal{L}$ from the product of spectrometer-derived cloud optical depth and drop effective radius (Eq. 3), as is typically done for MODIS-based retrievals, and have not addressed the question of how this derivation might affect the $S_o$ bias.

To provide context for the problem we start with a theoretical framework based on spatial distributions of the interacting aerosol and cloud fields, which helps identify key variables that control the $S_o$ bias (the variance in the fields and the correlation

between the fields). Then, picking up on earlier work (McComiskey and Feingold, 2012), which analyzed three stratocumulus cloud scenes with varying degrees of cloud field variance, we extend the analysis to 127 simulations exhibiting a wide variety of stratocumulus cloud scenes. The LES provides all essential microphysical properties from which standard retrievals can be performed. The key cloud field properties are cloud-top drop effective radius $r_e$, cloud optical depth $\tau$ (both of which are taken directly from the LES) and $A_c$ (based on LES $\tau$ and Eq. 4). $N_d$ is derived from Eq. (2) and $\mathcal{L}$ from Eq. (3). Within

the framework of standard regression analysis, we quantify the effects of aggregation on the variance in the fields and the correlation between the fields, and the implications for $S_o$ biases.

The assessment of spatial aggregation biases considers three methodologies. The first is standard level 2 (L2) satellite methodology, which retrieves cloud properties at high resolution (order 1 km) and aggregates them up to a scale on the order of 100 km. Given the limitations in our LES domain size (40 km), we instead compare $S_o$ based on 800 m and 6 km scales.

This forms the basis of our spatial scale aggregation. Recognizing that some might choose to use the more compact aggregated data as a start point, our second approach is level 3 (L3) methodology, which applies microphysical retrievals to spatially aggregated data. Considering that a key and uncertain variable in the $S_o$ calculation is the derived drop concentration ($N_d$), a third set of calculations repeats the level 2 analysis but uses the 'true' (LES-calculated) $N_d$, which we refer to as the L2$_N$ methodology. All consider perfect cloud albedo retrievals based on LES cloud optical depth (Eq. 4), removing the well-known cloud albedo aggregation bias from the discussion. Furthermore, derived variables $r_e$ and $\tau$, which are central to the $S_o$ bias





analysis have been assumed to be free of measurement error. Real-world satellite retrieval errors of these variables, especially in heterogeneous or broken cloud fields, could amplify (or perhaps counter) the biases identified here. Therefore, applying the conclusions directly to satellite studies should be done with caution.

Key results pertaining to spatial aggregation biases are

1. $S_o$ biases are generally positive for all three approaches and, consistent with theory, tend to increase with decreasing correlation between the fields (Fig. 4). For L2, biases can reach 500 %, whereas they are capped at about 100 % for L2$_\text{N}$. These biases will translate to biases in aerosol-cloud radiative forcing.

2. L3 methodology (falsely) creates negative correlations between the cloud fields, resulting in generally unpredictable $S_o$ bias behavior (Fig. 4b). Moreover, it generates negative $S_o$ values, exacerbated by increasing degrees of aggregation (Fig. 3b). The positive $S_o$ biases for L3 are therefore misleading (Figs. 3b and 4b).

3. High cloud fraction $f_c$ states are equally or even more prone to $S_o$ bias than low $f_c$ (high $D(\mathcal{L})$) states. This is in part due to the nature of the $N_d$ retrieval, which introduces heterogeneity into the field (see example Fig. 11), but L2$_\text{N}$ cases also tend to exhibit this trend. The literature tends to consider high $f_c$ states as homogeneous, but this works shows that this is not necessarily the case since the $N_d$ retrieval generates unrealistic small-scale heterogeneity in relatively homogeneous conditions.

4. Using regression theory (Eq. 10) we can interpret biases in $S_o$ based on the extent to which aggregation changes (i) the ratio of the correlation $r_{x,y}$ ($r$-ratio) and (ii) the ratio of $\sigma_y/\sigma_x$ ($\sigma$-ratio) at the 6 km vs the 800 m scales. Fig. 5 shows that that the $S_o$ bias is strongly dependent on the $\sigma$-ratio, particularly for L3 and L2$_\text{N}$, and for L2 at high $r(\mathcal{L}, N_d)$. The $r$-ratio is a weaker control over the $S_o$ biases (Fig. 6).

5. Since the $\sigma$-ratio is a ratio of ratios, we further expand our analysis of this term to assess the extent to which aggregation changes $\sigma_{A_c}$ relative to $\sigma_{N_d}$. We find that while aggregation can reduce $\sigma_{A_c}$ as much as $\sigma_{N_d}$, there is a tendency for larger reductions in $\sigma_{N_d}$ (Fig. 7), particularly for high $f_c$ and low $r$-ratio L2 cases (Fig. 8) and for low $f_c$ L2$_\text{N}$ cases (Fig. 7).

6. $S_o$ biases exceeding 100% always occur when aggregation-related smoothing has a stronger effect on $N_d$ than on $A_c$ (Fig. 9) although the magnitude and even sign of these biases vary significantly depending on the methodology. The lack of monotonicity in responses under these conditions depends on the relative strength of the $r$-ratio, and the extent to which it amplifies or counteracts the $\sigma$-ratio. This in turn, depends on the cloud fields themselves in ways that are not always easy to clarify.

7. As anticipated by Eq. (1) the $S_o$ bias is a strong function of the $\mathcal{L}_o$ bias (Fig. 10). In the case of L2, the $\mathcal{L}_o$ bias is noticeably smaller than the $S_o$ bias at low $f_c$, while the reverse is true for L3.

Regarding temporal aggregation, we note that if aerosol-cloud interaction metrics such as $S_o$ are based on aggregation of cloud scenes over an extended period of time, bias can be expected as a result of extending the range of conditions beyond the





natural local fluctuations inherent to covarying meteorological and aerosol conditions. To assess the effects of this temporal aggregation, we consider the difference between the average of $S_o$ from individual cloud scenes and the $S_o$ that is derived from

a best fit to the data from all those scenes, with the former reflecting the average of local-in-time $S_o$ and the latter reflecting the effect of temporal aggregation. Calculations are performed at both 800 m and 6 km scales (Table 1). Of note is that temporal averaging, as calculated in this manner, *reduces* (in % terms) $S_o$ at both averaging scales, and for L2, L3, and L2$_\mathrm{N}$ analysis. The negative bias is on the order of 70 – 80 % for L2 and L2$_\mathrm{N}$. As noted in section 4.1.1, L3 spatial aggregation methodology generates *negative $S_o$*, and temporal aggregation has the beneficial effect of creating small positive, and therefore more realistic

$S_o$ (relative to temporally unaggregated) values. (In percentage change terms, however, the bias is negative.)

We emphasize that these offsetting effects of spatial and temporal $S_o$ biases are very situationally-dependent and require further investigation. As in the case of the regression analysis (Eq. 10) performed for spatial aggregation, the pertinent question becomes the extent to which temporal aggregation will affect the $\sigma$- and $r$-ratios. Further work will need to place this offsetting effect on firmer footing.

Finally, it is of great importance that similar assessments of $S_o$ biases be considered in real-world satellite-based data. This will allow the community to assess the degree of coherence between the aforementioned studies and the model-based results presented here, and the implications for quantification of aerosol-cloud radiative forcing.





**Table 1.** Temporal averaging values and % differences. $\Sigma S_o$ refers to a simple average of the $S_o$ for individual cloudy scenes; $\bar{S}_o$ refers to the $S_o$ regression fit to the temporally aggregated data from all cloudy scenes; the % difference is defined as $(\bar{S}_o - \Sigma S_o)/\Sigma S_o$. $n = 4$ and $n = 30$ refer to the 800 m and 6 km spatial aggregation scales, respectively.

| | Temporal Averaging Values and % Differences | | | | | | | | |
| | L2 | | | L3 | | | L2$_\text{N}$ | | |
| | $\Sigma S_o$ | $\bar{S}_o$ | % | $\Sigma S_o$ | $\bar{S}_o$ | % | $\Sigma S_o$ | $\bar{S}_o$ | % |
| $n = 4$ | 0.564 | 0.259 | -74 | -0.652 | 0.044 | -107 | 1.012 | 0.301 | -70 |
| $n = 30$ | 0.966 | 0.298 | -69 | -0.959 | 0.073 | -108 | 1.373 | 0.296 | -78 |

*Data availability.*

Model output is available on request.

*Author contributions.* The study was conceived by GF. GF and TG designed the study. GF performed the analysis. Simulations were done by TY. All authors were engaged in reviewing results.

*Competing interests.* GF is an Associate Editor of ACP. The authors declare no other competing interests.

*Acknowledgements.* We thank the Department of Energy's Atmospheric System Research for supporting this work under Interagency Award 89243020SSC000055. T. G. acknowledges funding from the German Research Foundation Deutsche Forschungsgemeinschaft, DFG, project
"CDNC4ACI", GZ QU 311/27-1.





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





**Figure Captions**

Figure 1: Theoretical calculations of the albedo susceptibility bias for a range of relative dispersions in $\mathcal{L}$ and $N$ (D($\mathcal{L}$) and D($N$), respectively), for three different correlations ($r$) between $\mathcal{L}$ and $N$. Note the values of large positive and negative biases.

Figure 2: As in Fig. 4 but for the $S_o$ bias as a function of the spatial correlation between $\mathcal{L}$ and $N$, for a range of relative
dispersions in $\mathcal{L}$ (D($\mathcal{L}$)) and fixed D($N$). Of note is that for the conditions shown the biases are large and range from about
-150 % (for positive $r$ and large D($\mathcal{L}$)) to + 70 % (negative $r$, large D($\mathcal{L}$)).

Figure 3: $S_o$ aggregated to a 6 km scale ($\bar{S}_o$) vs. $S_o$ aggregated to an 800 m scale ($S_o$) for (a) L2, (b) L3 and (c) L2$_\mathrm{N}$
methodology as described in the text. Solid line is the 1:1 line. Dashed lines are drawn at ordinate and abscissa values of zero.
Note the general overestimate in $S_o$ with increasing aggregation scale in (a) and (c) and the change of sign in $S_o$ associated
with L3 aggregation in (b). Of note is that high cloud fraction/low heterogeneity conditions are often associated with high
biases.

Figure 4: $S_o$ bias in % as defined in Eq. 11 as a function of the correlation between liquid water path and drop concentration
$r(\mathcal{L}, N_d)$ with points colored by cloud fraction $f_c$ for (a) L2, (b) L3 and (c) L2$_\mathrm{N}$ methodology. In (a) and (c) the $S_o$ bias tends
to increase with decreasing $r(\mathcal{L}, N_d)$ and increasing cloud fraction $f_c$, with noticeable exceptions. Use of the true $N_d$ in (c)
restricts the $S_o$ bias to about 100%.

Figure 5: As in Fig. 4 but with points colored by the ratio of $\sigma_{A_c}/\sigma_{N_d}$ at 6 km to $\sigma_{A_c}/\sigma_{N_d}$ at 800 m (the $\sigma$-ratio). Note the
clear dependence of the $S_o$ bias on the $\sigma$-ratio and that the bias $\sim 0$ when the $\sigma$-ratio $\sim 1$ (green colors). Exceptions to the latter
occur for L2 at low $r(\mathcal{L}, N_d)$.

Figure 6: As in Fig. 4 but with points colored by the ratio of correlations $r(\mathcal{L}, N_d)$ at 6 km to $r(\mathcal{L}, N_d)$ at 800 m (the $r$-ratio).
Comparison with Fig. 5 shows approximately orthogonal dependence of the $\sigma$- and $r$-ratios on the $S_o$ bias. In (a) and (b),
calculation of the $r$-ratio is mathematically unstable at $r(\mathcal{L}, N_d) = 0$ as evidenced by saturating colors.

Figure 7: Effects of aggregation on normalized reduction in $\sigma(A_c)$ vs. normalized reduction $\sigma(N_d)$ to shed light on results in
Fig. 5. Points are colored by $f_c$. At high (low) $f_c$ aggregation tends to more significantly reduce (increase) $\sigma(N_d)$ relative to
$\sigma(A_c)$. Notable exceptions occur in (c) for some low $f_c$ conditions. See text for further discussion.

Figure 8: As in Fig. 7 but with points colored by the $r$-ratio. See text for discussion.

Figure 9: As in Fig. 7 but with points colored by $S_o$ bias. Biases exceeding 100 % always occur when aggregation-related
smoothing has a stronger effect on $N_d$ than on $A_c$.





Figure 10: $S_o$ bias vs. the $\mathcal{L}$-adjustment bias ($\mathrm{dln}\mathcal{L}/\mathrm{dln}N_d$ in Eq. 1) for a limited set of model scenes (for clarity). Note a strong positive correlation between the two biases, as expected from Eq. (1). In (a) and (b), the relative magnitude of these biases depends on $f_c$, although with opposite trend. In (a) and (c), the $\mathcal{L}$-adjustment bias tends to be larger than the $S_o$ bias.

Figure 11: High resolution ($n = 1$) 2-D snapshot of (a) 'true' $N_d$, (b) retrieved $N_d$ (Eq. 2); (c) $\mathcal{L}$, and (d) the relationship between (b) and (a). Note the different scales between (a) and (b). Although the mean values of $N_d$ in (a) and (b) differ by only -36 %, they exhibit negative correlation over the scene. The retrieved $N_d$ introduces significant heterogeneity into the field. Use of the true $N_d$ reduces the absolute value of the $S_o$ error from -2017 % to +80 %.

Figure 12: As in Fig. 11 but for an open-cellular case. Note the different scales between (a) and (b). Mean values of $N_d$ in (a) and (b) differ by -75 %. Use of the true $N_d$ degrades the $S_o$ bias (285 % for L2 vs. 803 % for L2$_\mathrm{N}$).





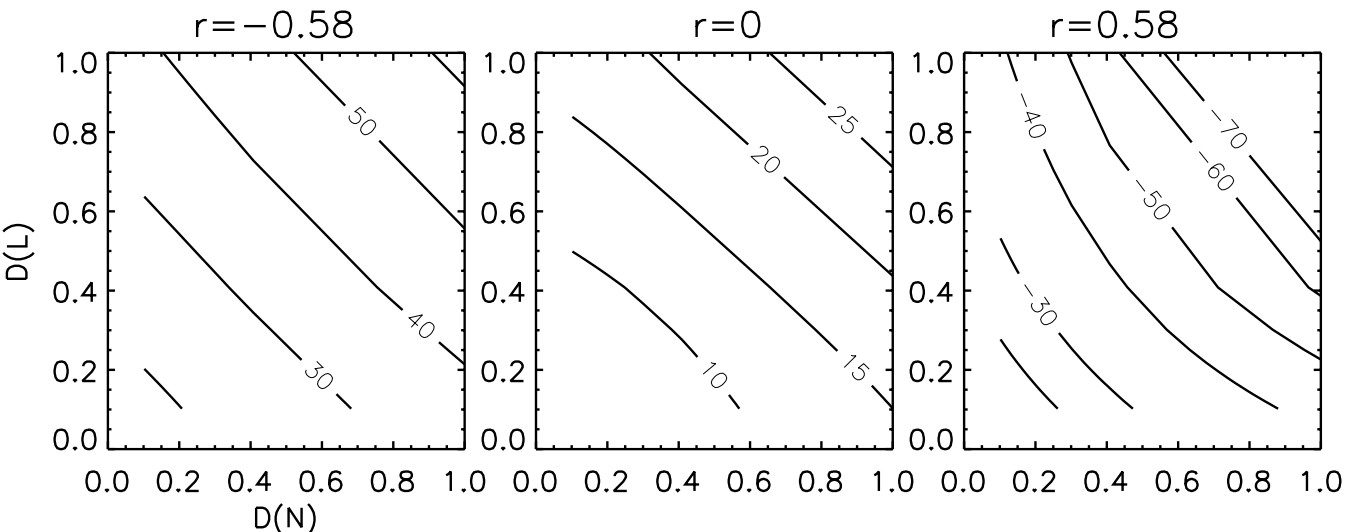

Figure 1: Theoretical calculations of the albedo susceptibility bias for a range of relative dispersions in *L* and *N* (*D(L)* and *D(N)*, respectively), for three different correlations (*r*) between *L* and *N*. Note the values of large positive and negative biases.



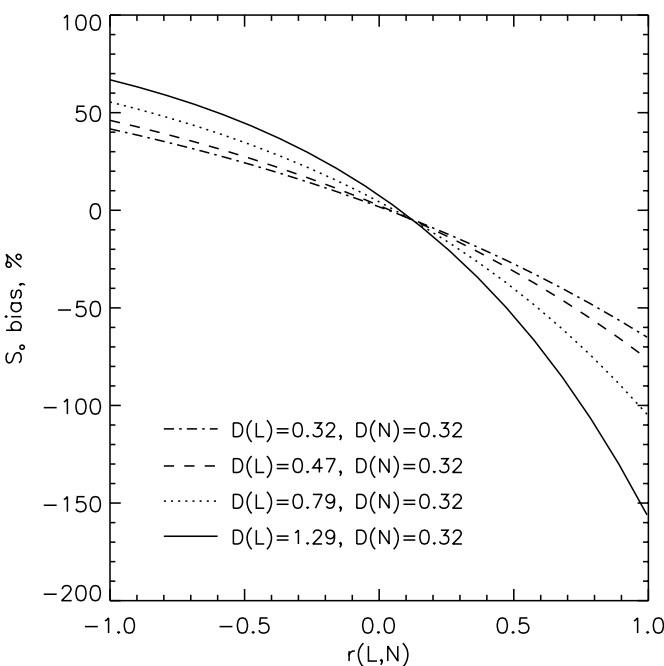

Figure 2: As in Fig. 1 but for the $S_o$ bias as a function of the spatial correlation between $L$ and $N$, for a range of relative dispersions in $L$ ($D(L)$) and fixed $D(N)$. Of note is that for the conditions shown the biases are large and range from about -150% (for positive $r$ and large $D(L)$) to + 70% (negative $r$, large $D(L)$).





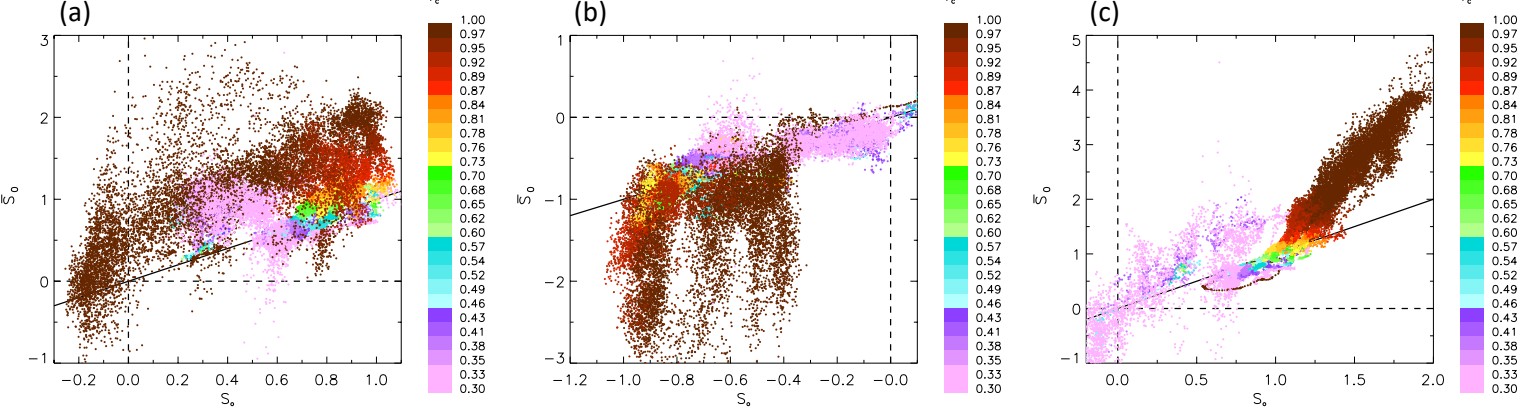

Fig. 3: $S_o$ aggregated to a 6 km scale vs. $S_o$ aggregated to an 800 m scale for (a) L2, (b) L3 and (c) L2$_N$ methodology as described in the text. Solid line is the 1:1 line. Dashed lines are drawn at ordinate and abscissa values of zero. Note the general overestimate in $S_o$ with increasing aggregation scale in (a) and (c) and the change of sign in $S_o$ associated with L3 aggregation in (b). Of note is that high cloud fraction/low heterogeneity conditions are often associated with high biases.



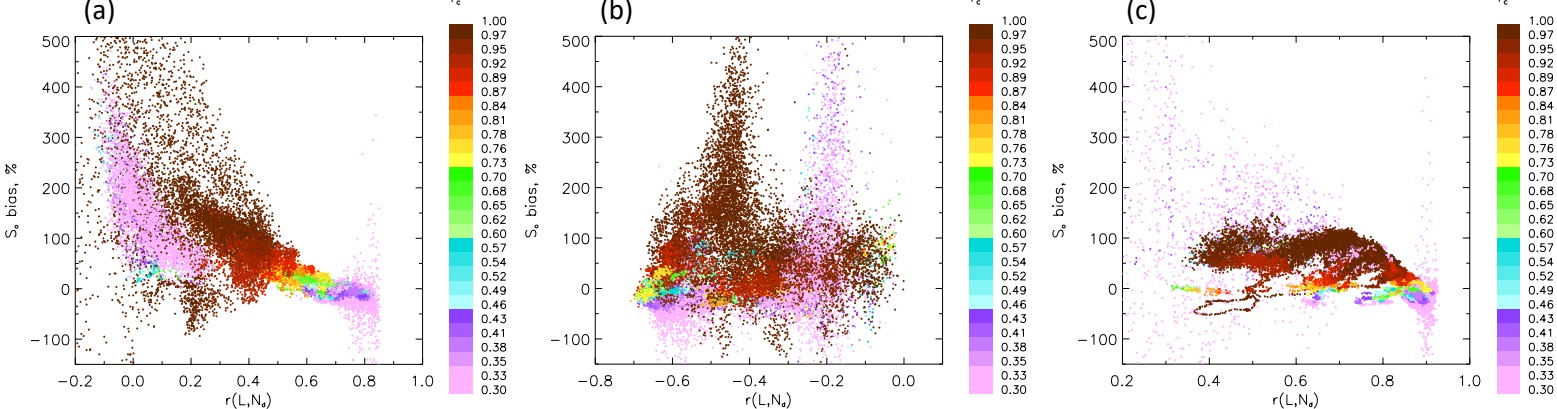

Fig. 4: $S_o$ bias in % as defined in Eq. 11 as a function of the correlation between liquid water path and drop concentration $r(L, N_d)$ with points colored by cloud fraction $f_c$ for (a) L2, (b) L3 and (c) L2$_N$ methodology. In (a) and (c) the $S_o$ bias tends to increase with decreasing $r(L, N_d)$ and increasing cloud fraction $f_c$, with noticeable exceptions. Use of the true $N_d$ in (c) restricts the $S_o$ bias to about 100%.





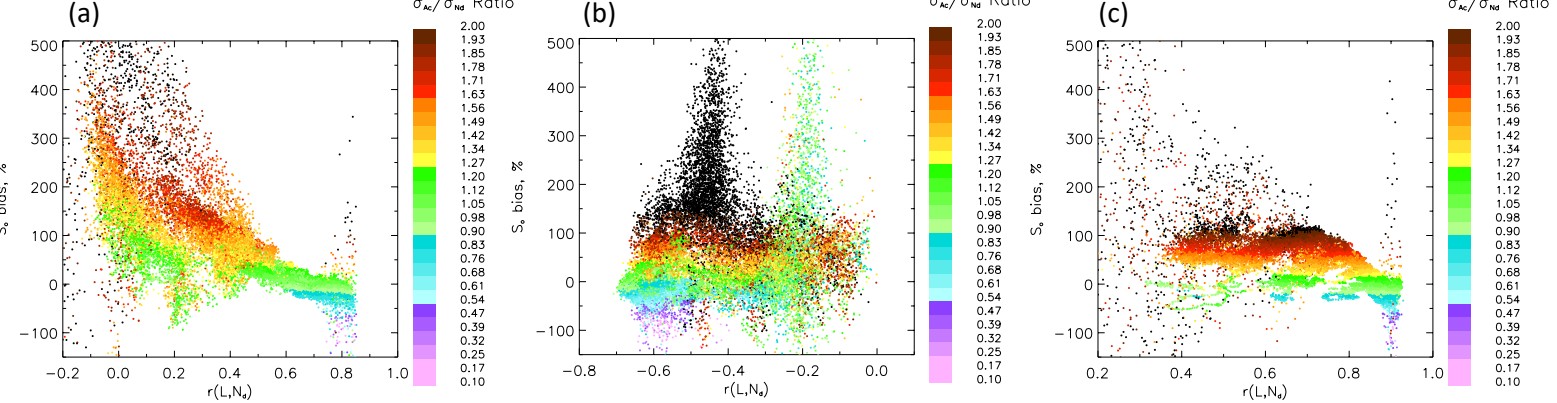

Fig. 5: As in Fig. 4 but with points colored by the ratio of $\sigma_{Ac}/\sigma_{Nd}$ at 6 km *to* $\sigma_{Ac}/\sigma_{Nd}$ at 800 m (the $\sigma$ - ratio). Note the clear dependence of the $S_o$ bias on the $\sigma$ - ratio and that the bias ~ 0 when the $\sigma$ - ratio ~1 (green colors). Exceptions to the latter occur for L2 at low $r(L, N_d)$.



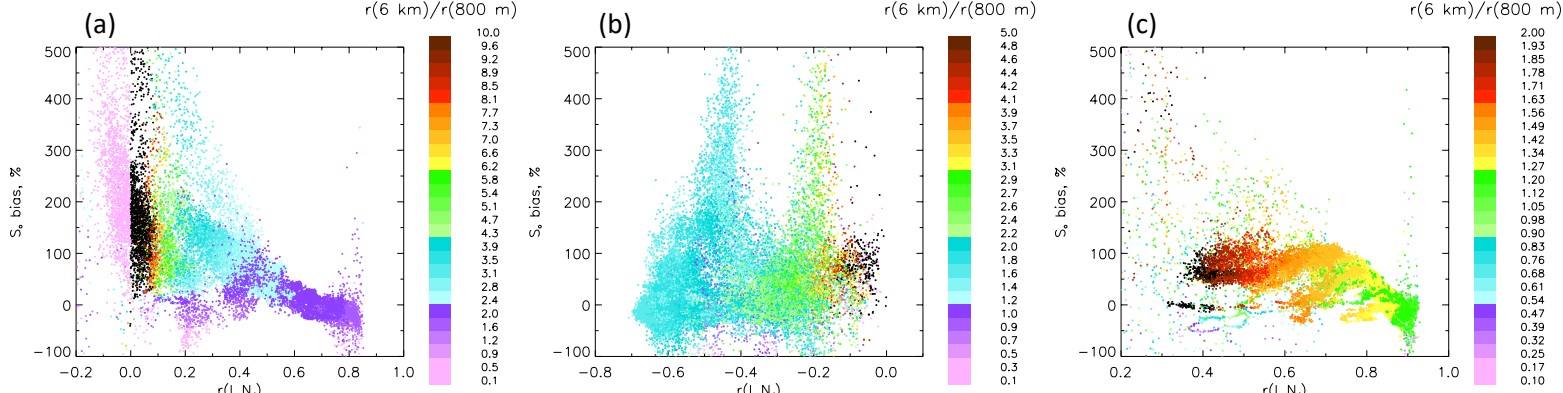

Fig. 6: As in Fig. 4 but with points colored by the ratio of correlations $r(L, N_d)$ at 6 km to $r(L, N_d)$ at 800 m (the $r$-ratio). Comparison with Fig. 5 shows approximately orthogonal dependence of the $\sigma$- and $r$- ratios on the $S_o$ bias. In (a) and (b), calculation of the $r$-ratio is mathematically unstable at $r(L, N_d)$ =0 as evidenced by saturating colors.





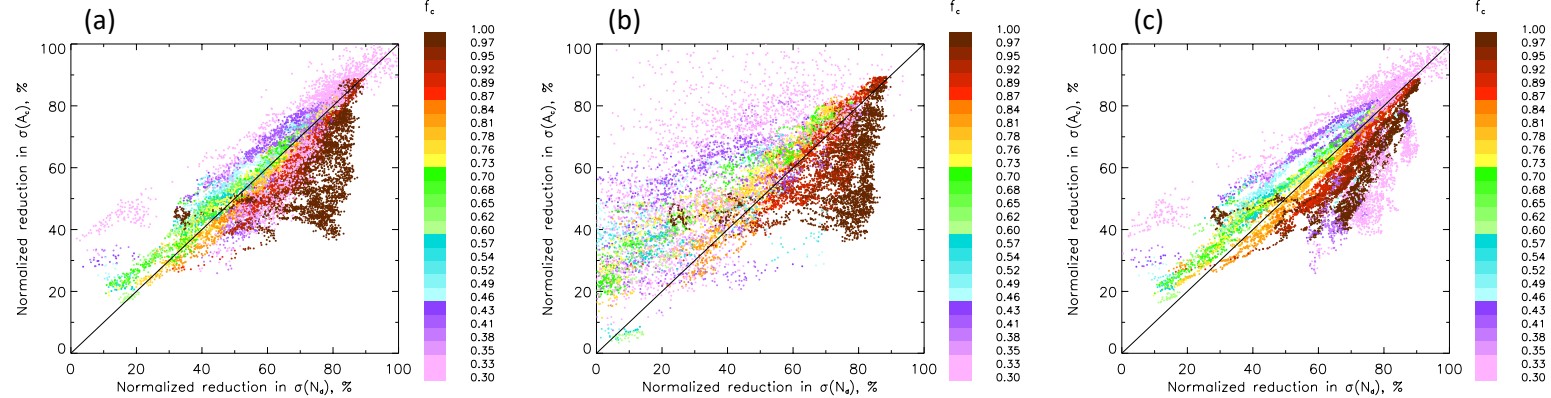

Fig. 7: Effects of aggregation on normalized reduction in $\sigma(A_c)$ vs. normalized reduction $\sigma(N_d)$ to shed light on results in Fig. 5. Points are colored by $f_c$. At high (low) $f_c$ aggregation tends to more significantly reduce (increase) $\sigma(N_d)$ relative to $\sigma(A_c)$. Notable exceptions occur in (c) for some low $f_c$ conditions. See text for further discussion.



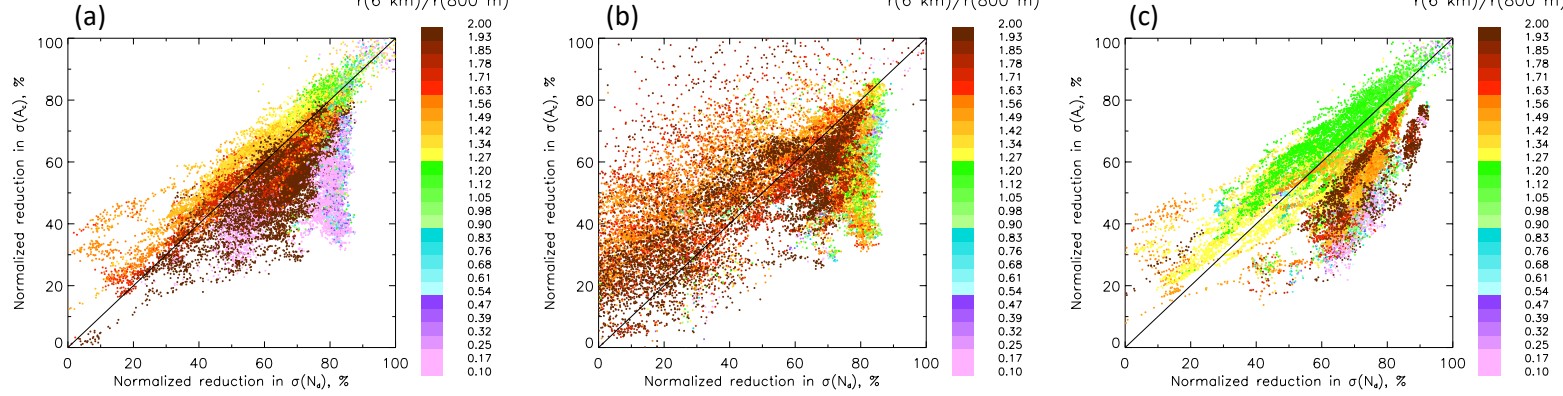

Fig. 8: As in Fig. 7 but with points colored by the *r*-ratio. See text for discussion.





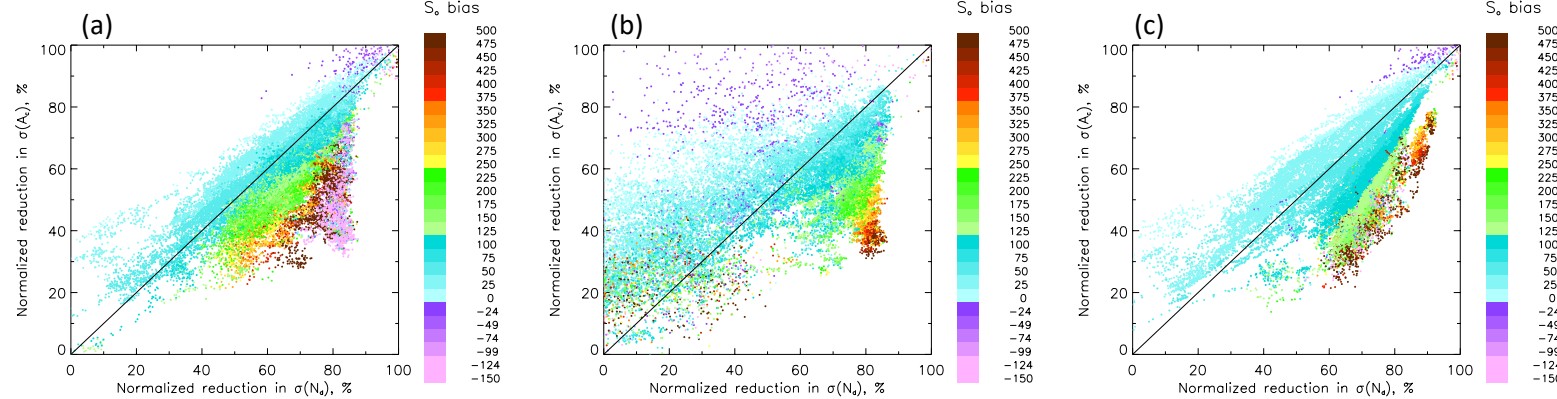

Fig. 9: As in Fig. 7 but with points colored by $S_o$ bias. Biases exceeding 100 % always occur when aggregation-related smoothing has a stronger effect on $N_d$ than on $A_c$.





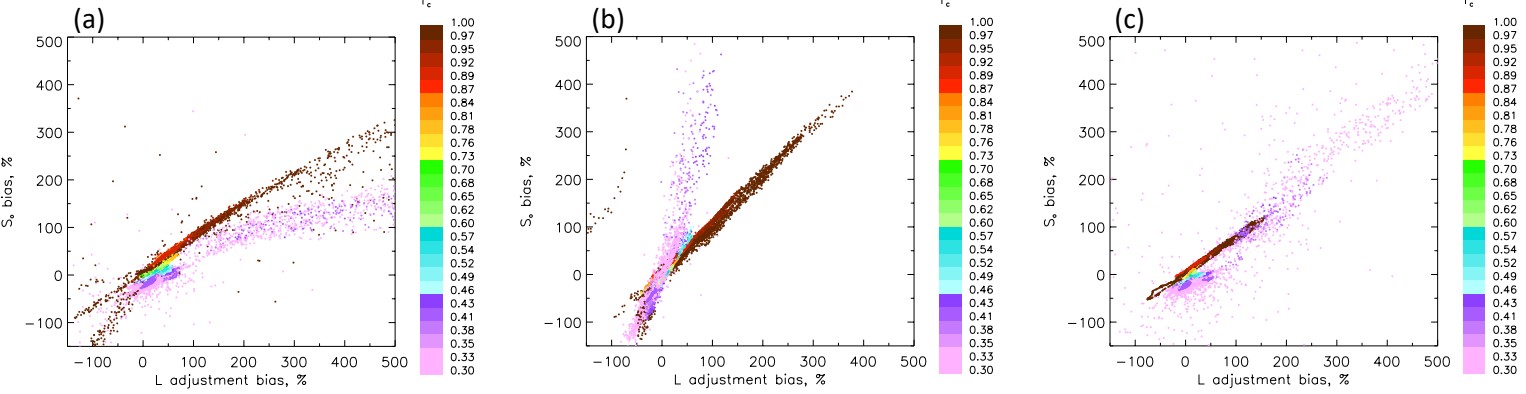

Fig. 10: $S_o$ bias vs. the $L$-adjustment bias (dln $L$/dln $N_d$ in Eq. 1) for a limited set of model scenes (for clarity). Note a strong positive correlation between the two biases, as expected from Eq. (1). In (a) and (b), the relative magnitude of these biases depends on $f_c$, although with opposite trend. In (a) and (c), the $L$-adjustment bias tends to be larger than the $S_o$ bias.

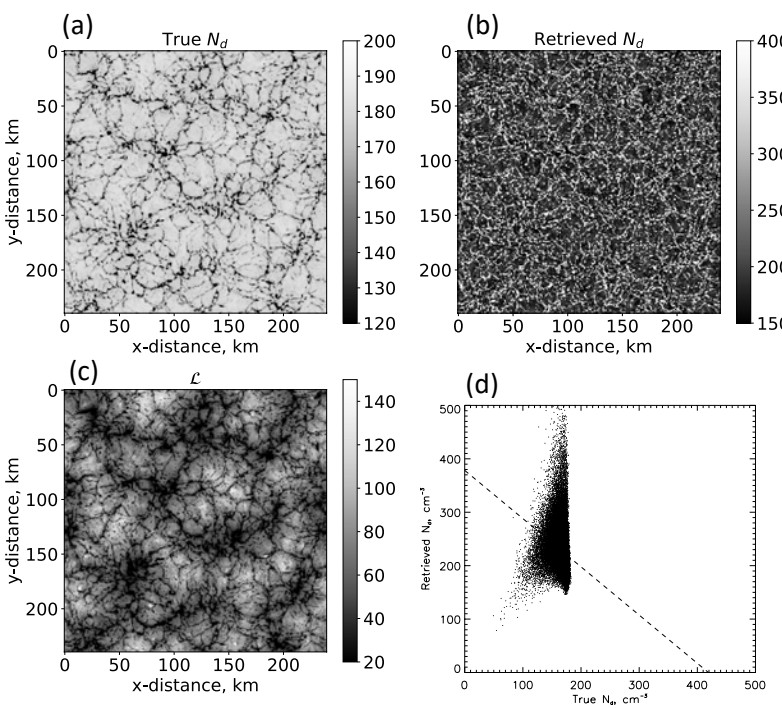

Fig. 11: High resolution ($n =$1) 2-D snapshot of (a) 'true' $N_d$, (b) retrieved $N_d$ (Eq. 2); (c) $L$, and (d) the relationship between (b) and (a). Note the different scales between (a) and (b). Although the mean values of $N_d$ in (a) and (b) differ by only -36 %, they exhibit negative correlation over the scene. The retrieved $N_d$ introduces significant heterogeneity into the field. Use of the true $N_d$ reduces the the absolute value of the $S_o$ error from -2017% to +80 %.



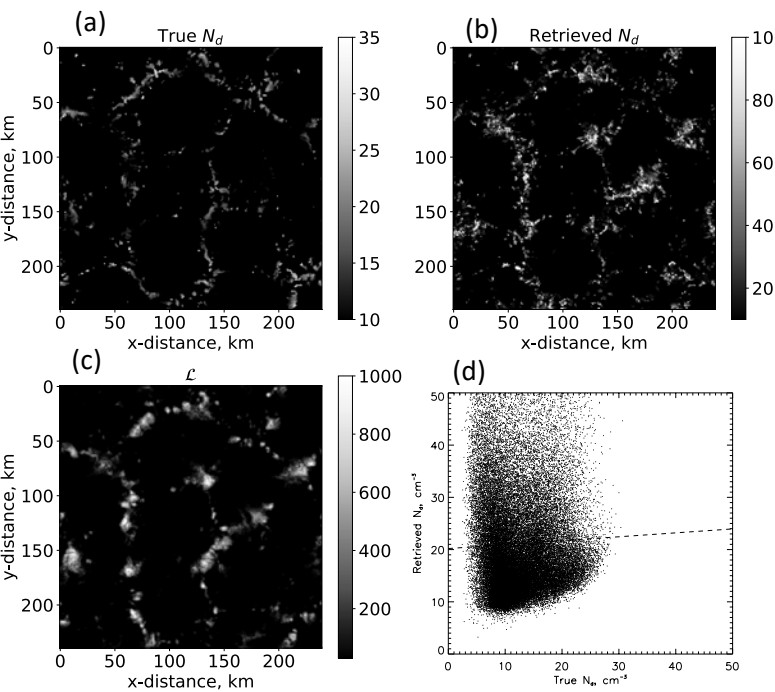

Fig. 12: As in Fig. 11 but for an open-cellular case. Note the different scales between (a) and (b). Mean values of $N_d$ in (a) and (b) differ by -75 %. Use of the true $N_d$ degrades the So bias (285 % for L2 vs. 803 % for L2N).