# Peer review of "Quantifying Albedo Susceptibility Biases in Shallow Clouds"

_Atmospheric Chemistry and Physics, 2021_

## Author Comment (AC1)

**Overall comment**

This study explores roles of spatial and temporal aggregation on quantifying albedo susceptibility ($S_o$) biases by analyzing the outputs of an ensemble of 127 large eddy simulations of marine stratocumulus. The authors designed three methodologies (L2, L3, and $L2_N$), which mimic common satellite-based analyses in different ways, to identify the influences of the adiabatic drop concentration Nd retrieval, the correlation between aerosol and cloud fields, and the extent of reduced variance in cloud albedo and Nd. The LES simulations also provide an opportunity to obtain the 'true' Nd, by which the effectiveness of adiabatic assumption used in satellite retrievals can be assessed. As a consequence of such an analysis, the authors obtained a series of interesting results regarding the influencing factors on albedo susceptibility biases. I think this is a very nice study, and the results presented also have significant implications for reconciling currently diverse observation-based estimates of aerosol indirect effects.

I would recommend this paper be published in Atmospheric Chemistry and Physics after my specific questions/concerns listed below are addressed appropriately.

Thank you for the careful review. Text additions are in blue and changes in red.

**Specific comments**

*Line 106-111: If I understand correctly, B (in Eq. 9) is only relevant to the sensitivity of L to Nd, i.e., dln(L)/dln(Nd), not to the whole albedo sensitivity. It's a bit confusing for me how authors translate B to the theoretical calculation of the $S_o$ biases?*

Thanks for this important comment. The calculation was not explained. We revisit the methodology here:

1) We assumed for simplicity that $S_o \sim (1-A)/3$ (i.e., no L adjustments)
2) We assumed $A \sim \tau$ (reasonable at low enough $\tau$)
3) Our two interacting fields are L and N
4) Therefore $\tau \sim L^{\wedge}(5/6) N^{\wedge}(1/3)$
5) We calculated $B_x$ and $B_y$ (where x=L and y=N, $\beta_L$ =5/6, $\beta_N$=1/3) and $B_{cov}$
6) Finally, the overall bias was calculated according to Eq. (9)

We do note that this is simplified, and representative of interacting L and N with their co-variabilities and spatial heterogeneities. When embarking on the LES-based calculations, we use $\tau$ and $r_e$ as our key variables (to conform with the satellite approach), and L and N are derived from $\tau$ and $r_e$. In this sense our Fig. 1 calculations are distinct from the LES calculations and are therefore mostly illustrative. See text additions on lines 115-121.

*Line 128: As for the LES, why did the authors only choose "nocturnal" instead of "diurnal" simulations or both?*

We had performed a large set of simulations for a previous study and since these were readily available, we chose to use them. The important point is that they represent many different cloud scenes, ranging from closed- to open-cells, and a range of boundary layer/cloud depths, and cell sizes.

*Line 139: How did the authors determine γ value in the calculation of Ac from simulated cloud optical depth?*

As noted in the original manuscript on line 139 (now lines 157-158), $A_C$ is calculated based on the modeled value of $\tau$ (Eq. 4).

*Line 184: At what spatial resolution is cloud fraction defined here? Is it at 48km x 48km, or defined at 800m and 6 km respectively and then averaged up for whole domain?*

Cloud fraction is defined at the native resolution since we're using it as a measure of the cloud field property. This is now clarified on line 204.

*Line 186: It is expected that high fc (homogeneous clouds) would be associated with low bias in S. Why is the opposite here?*

The way we chose to write this was as a process of discovery. One has to dig more deeply into the analysis before one can answer this question. But we have now added text to explain that the reason for this unexpected result will be revealed later (line 207). An easy, quick explanation is seen in Fig. 11. The adiabatic Nd retrieval in this high cloud fraction scene generates far too much heterogeneity, which results in strong $S_o$ bias.

*Line 184-192: The comparison between L2 and L3 methods here is to illustrate the aggregation biases associated with Jensen's inequality. Actually, there is already another practical method accounting for this issue based on satellite observations. For example, the MODIS L3 product includes a cloud optical depth-effective radius joint histogram which was suggested to consider the non-linearity in the calculation of Nd (e.g. Quaas et al., 2008; Grandey and Stier, 2010). Thus, it might be interesting to evaluate the effectiveness of this method from the LES data in this study.*

Thank you for bringing these studies to our attention. The dependence of ACI metrics on scale in Grandey and Stier (2010) is relevant, although we note that they used level 3 data and aggregation scales ranging from 1-degree to 4-degree to 60-degree boxes. We now cite this paper in the revised text (line 80-83) but do not engage further because we don't see a straightforward way of connecting to our paper without major digression.

*Line 274: How did the authors select these 58 simulations? Is there an objective criterion? Will it introduce artificial selection on cloud regimes? To show the robustness, it is useful to present the results from all 127 simulations, at least in the supplement information.*

Since the 58 simulations are Latin Hypercube samples of the total, they are not biased. For example, the figure shows the full 127 simulations at left compared to the subsample at right. Because the points tend to align with a clear linear relationship, they quickly obscure points below, particularly at the intermediate cloud fractions. Text has been added on line 297-298.

[Figure]

*Line 280: It's interesting that the separation of these branches for L2$_N$ is not as evident as L2 and L3. What is the underlying reason? The authors should explain in more detail.*

The reason is that with the correct Nd, L2$_N$ avoids the bias and variance in Nd associated with clouds that have morphological structure. It is not only broken clouds but all clouds for which Nd retrievals generate unrealistic structure. This is now stated in the revised text on lines 306-307.

*Line 300: Generally, a negative bias in retrieved Nd is expected duo to a positive bias in CER and a negative bias in COT for spatially inhomogeneous scenes according to the Eq.2 (Grosvenor et al., 2018). Thus, it is kind of surprising that the retrieved Nd for open-cellular clouds is larger than the true Nd.*

Remember that we are assuming that $r_e$ and $\tau$ are taken directly from the model, with no assumptions. In other words, we are not dealing with the broken cloud $r_e$ and $\tau$ biases. We are only focusing on the effect of averaging in clouds with different morphologies. We now clarify this in the text on lines 333-338.

*Line 317: It is not quite clear that how the authors conducted the regression fits. As for ∑S$_o$, does the 'individual scenes' here mean the whole domain? In this case, the regression fit was conducted over all 4x4- (or 30x30-) resolved grids in each scene, and then ∑S$_o$ was calculated by averaging up the individual S over all scenes (including the variations along both time and different simulations). Please clarify more detail on how the authors conducted the analysis.*

The ∑S$_o$ is calculated as follows: first each scene (i.e., the entire domain) is used to calculate the sum; then the individual So values are averaged. This is done for all scenes that meet the criteria discussed in Section 3, and applied to all the other analyses. The second approach includes the same data, but now the regression is performed on all the aggregated data. In other words, the relationship now removes the natural co-variability associated with individual scenes. We now explain this more carefully in the revised text on lines 349-351.

Grandey, B. S., and P. Stier (2010), A critical look at spatial scale choices in satellite-based aerosol indirect effect studies, Atmos. Chem. Phys., 10(23), 11459–11470, doi:10.5194/acp-10-11459-2010.

Grosvenor, D. P. et al. Remote sensing of cloud droplet number concentration in warm clouds: a review of the current state of knowledge and perspectives. Rev. Geophys. 56, 409–453 (2018)

Quaas, J., O. Boucher, N. Bellouin, and S. Kinne (2008), Satellite-based estimate of the direct and indirect aerosol climate forcing, J. Geophys. Res., 113, 05204, doi:10.1029/2007JD008962.

[Figure]

Reply

---

## Author Comment (AC2)

**RC2**: 'Comment on acp-2021-859', Anonymous Referee #2, 21 Dec 2021  reply
General comment:

This study analyzes output of LES simulations for marine stratocumulus clouds to investigate how small-scale cloud variables and their relationships to aerosols are aggregated to manifest the albedo susceptibility bias occurring at larger scales typical of satellite-based analysis with L2 and L3 datasets. For this purpose, theoretical relationships between key statistical properties are reviewed and applied to the LES output to quantify the albedo susceptibility bias as a function of several statistical properties and cloud water susceptibility for different aggregation scales. This study offers a nice demonstration of how spatial cloud inhomogeneity and non-linear aerosol-cloud relationships are a source of uncertainty in quantifying the albedo susceptibility, directly relevant to radiative forcing due to aerosol-cloud interaction. I have some minor comments mostly regarding the presentation and/or clarification as listed below and I would recommend the manuscript to be considered for publication in ACP after the authors properly address the comments.

We thank the reviewer for raising interesting questions, which have resulted in the addition of two appendices. Text additions are in blue and changes in red.

Specific comment:

*Section 3.1: It is hard (at least for me) to understand in detail how the LES output variables (at the native model grid resolution) are averaged and/or aggregated into different spatial scales. In particular, I am confused with the term "aggregated" which seems to mean "averaged" for some parts and to mean "accumulated" for other parts. I would appreciate the authors to clarify if each "aggregated" means "averaged" or "accumulated". Please look at the Minor Points listed below for specific locations in the text for this clarification.*

Thank you for this comment. In this work, the terms 'aggregated' and 'averaged' mean the same thing. We tend to use 'aggregated' when we speak more broadly about including data from a larger range of spatial and temporal scales. We now clarify this in the text on lines 38-40.

*Line 188-190: "This is because L3 averaging has a stronger smoothing effect on broken cloud fields, and therefore somewhat unexpectedly reduces the averaging bias for broken cloud fields compared to solid cloud fields": Does this explain the negative values of the albedo susceptibility in Fig. 3b?*

This is an interesting point that we believe is a consequence of the Simpson paradox, a simple example of which is given below and included in Appendix A. In the example below, we expect basketball success to generally increase with a player's height. This is indeed the case when stratifying the data (in this case by age-group), i.e., **avoiding aggregation**. But when one averages the red points to a single value, and similarly, averages the blue points to a single value gives an *apparent negative relationship* between ordinate and abscissa. See Appendix A.

[Figure]

*Section 4.1.3: The argument here associated with Fig. 10 is of particular importance in the context of cloud water adjustment and its impact on albedo. It is interesting to see that the tight correlation between S0 and L adjustment biases relates inversely with cloud fraction between L2 and L3. How can this reversed relation be understood? Please add some more discussion.*

In response to this question, we have dug a little deeper into the model output. Based on Fig. 7 and Eq. (13), our intuition was that the change in the slope with respect to cloud fraction between L2 and L3 in Fig. 10 is likely a function of the change in aggregation-smoothing in $A_c$ vs. smoothing in L. To test this, we repeat Fig. 7 (top row), but now also look at smoothing in L (bottom row):

[Figure]

What is apparent is that for L3, there is much more smoothing in $A_c$ than in L at low cloud fractions (the L3 points lie closer to the 1:1 line in (e) vs. (b) for low cloud fraction, $f_c$). For L2, more low $f_c$ points tend to lie below the 1:1 line in (d) vs (a) but because of this migration of points from above to below the line, it is more difficult to interpret. For L2N the smoothing in L and $A_c$, look fairly similar, in line with our intuition that the $S_o$ bias vs. L adjustment bias (Fig. 10) is related to the aggregation-smoothing of $A_c$ vs the aggregation-smoothing of L (cf Fig. 10 where one sees less of a bias in the relationship for low and high cloud fraction points).

To dig even deeper, we looked more closely at the sigma-ratio (Eq. 12) for the low cloud fractions. We fit a linear relation between points for $0.3 < f_c < 0.43$ (shown on the plot below) and $f_c > 0.85$ (not shown) and obtained the following:

$0.3 < fc < 0.43$                                                    $fc > 0.85$

L2: y = 0.06 + 0.78 x                                 L2: y = 0.1 +0.90 x

L3: y = -0.25 + 1.03 x                               L3: y = 0.04 + 0.94 x

L2N: y =-0.22 + 1.02 x                            L2N: y = 0.23 + 0.81 x

The high $f_c$ slopes between L2 and L3 are similar (0.90 vs. 0.94, resp.) but there is a big difference in the low $f_c$ slopes (0.78 vs. 1.03 for L2 and L3 resp.), and in the direction consistent with Fig. 10ab. Note too, the negative intercept for L3 at low $f_c$, which is consistent with Fig. 10b.

[Figure]

Differences are even clearer if one zooms in on a log-log scale to accentuate the low $f_c$ points. (Note the linear fit is still applied but becomes curved on the log-log plot.):

[Figure]

Note the steepening of the low $f_c$ points in L3. Comparing with Fig. 10 confirms that it is the differences in degree of aggregation-related smoothing between $A_c$ and L in these low cloud fraction scenes that flips the relative sign of $S_o$ vs L adjustment slopes between Fig. 10a and Fig. 10b. We have added text to the revised document (line 303-304) to explain this and added supporting figures in Appendix B

We note that these differences in smoothing derive from the derivations of $A_c$ (Eq. 4) and L (Eq. 3), with $A_c$ a (non-linear) function of $\tau$ only, and L a function of the product of $\tau$ and $r_e$. Anticipating how the aggregation biases play out is not intuitive, and requires, in our experience, analyses of the kind shown here.

Minor point:

*Equation 6: Is this $B_x$ inverse of the bias?*

$B_x$ is related to the bias that occurs when smoothing small scale structure. It may be an enhancement factor if x < 1 or a reduction factor if x > 1. See, e.g., https://doi.org/10.5194/acp-19-1077-2019, Eq. (14)

*Fig. 1: Are the numbers labelled for contours in percent?*

Yes, corrected.

*Line 135: Insert 'of' between 'fields' and 'drop'*

Corrected.

*Line 138: "Both cloud water and rain water contribute to tau and re": Does this mean that re is computed as the ratio of third to second moments of the whole range of the bimodal size distribution? If so, is it consistent with what satellite measurement looks at given satellite measurement is sensitive primarily to the cloud mode alone?*

We include both cloud and rainwater since by definition, $r_e$ is the ratio of volume to surface area, and because rain water can on occasion contribute significantly to moments of the drop size distribution. A more rigorous calculation based on a MODIS instrument simulator might reveal some differences

but we believe they would be small, and that we are capturing the most important characteristics of the system.

*Line 146: "aggregated": Does this mean "averaged"?*

See response to specific point above and lines 38-40.

*Line 151: "at the pixel level": Does this mean the model native resolution (200m)?*

Yes, clarified on line 204.

Line 162: "aggregated": Does this mean "averaged"? Namely, are tau and re first averaged to n=4 and n=30, and then Equations (2) and (3) are applied to them to derive Nc and L?

Yes, correct.

*Line 201: Fig. 3 -> Fig. 4 (?)*

Yes, thank you for catching this mistake. See line 221.

*Line 206: Does this "aggregate" mean "average"?*

Yes, see response to specific point above and lines 38-40.

*Line 214: Please state that b(6km) and b(800m) are denoted as b_hat_overbar and b_hat, respectively.*

Done. (line 237)

*Fig. 3 etc.: Please put the panel titles as "L2", "L3" and "L2N".*

Done.

---

## Author Comment (AC3)

**RC3: 'Comment on acp-2021-859', Anonymous Referee #3, 23 Dec 2021 reply General**

Based on large-eddy simulations of marine cloud fields off California, this study explores the effects that spatial and temporal averaging as explicitly and implicitly done in satellite data analysis can have on the study of aerosol--cloud--climate interactions. The authors present a careful and well-documented analysis that puts existing satellitebased studies into perspective. I think this is a strong paper already, but have a few remarks mostly concerning methodology.

Thank you for the careful review. Text additions are in blue and changes in red.

**Details**

• *line 10: which biases? Bias of aggregation vs. inndividual models, or the other way around?*

We have modified the text to make it clear that we refer to the  $S_o$  bias (line 10).

• line 11: Explain L

Thank you. This was an omission. We now use liquid water path and define L on first usage.

• line 49: 'known'

Thank you. Corrected on line 52

• line 89: 'interest'

Thank you. Corrected on line 96

• line 93: 'well-known'

Thank you. Corrected on line 66

• line 96: 'the' standard deviation

Thank you. Corrected line 103.

• line 138: For 'cloud top', do you use the top-most layer of the model, or do you allow for some radiation penetration into the cloud, as found in satellite retrievals at smaller MIR wavelengths?

The cloud top  $r_e$  is calculated based on a liquid water mixing ratio threshold (0.01 g/kg). Because these clouds are strongly capped, the first grid point exceeding this value (when working downward towards the cloud top) almost always exceeds the threshold by a lot. Visual inspection of the data persuaded us that the values were as expected. In other words, we mimic a satellite retrieval in the sense that we use cloud-top  $r_e$ , but don't apply a simulator. We now make this clear in the text on lines 155-157.

• *line 146: In the aggregation, did you consider partial cloudiness in your cells as a weighting factor, i.e. via horizontal cloud fraction?*

Aggregation to 800 m is performed using a simple box-averaging algorithm, since this is essentially what a satellite-based instrument with that sensor resolution would see. To be consistent with the 800 m averaging, we do not perform any weighting when we average to 6 km. Again, we apply the box-average at the 6 km scale.

 line 148: Why did you choose 800mx800m to get 'close to the typical 1 km ...' of MODIS instead of using n=5 directly?

Thanks for this question. We have an even number of points in the domain and so we chose to use an even number of points in the box-average. This is now mentioned in the text on lines 168-169.

• Figures 11 and 12: Harmonizing the color bars on a and b would make it easier to compare both - assuming that any spatial detail was retained in so doing

Because the ranges differ significantly, we found that a great deal of dynamic range was lost. This was in part the reason for panel (d)

• *line 289: (sub)adiabatically*

Thank you. Corrected. (line 315)

• *lines 376--380: This is quite substantial, and the result that most surprised me. Based on this insight, do you have suggestions on how to improve Nd retrievals?*

The typical approach to this problem is to average over fairly large areas. This tends to produce much better results. But as this paper has shown, there are consequences to this averaging with respect to calculation of 'slopes'. Based on this work, we have an idea under development that we hope will improve  $N_d$  retrievals and susceptibility calculations. The goal is to test the idea on LES output and then apply it to satellite-based data. However, it is still in its development phase and not ready to be shared.